# Conjoined Twins Complicating a Dichorionic Triplet Pregnancy after Intracytoplasmic Sperm Injection: A Case Report and Review of the Literature

**DOI:** 10.3390/children9101549

**Published:** 2022-10-12

**Authors:** Anna Eleftheriades, Panagiotis Christopoulos, Elsa Tsapakis, Ermioni Tsarna, Nikolaos F. Vlahos, Emmanouil Kalampokas, Daniele Bolla, Makarios Eleftheriades

**Affiliations:** 1Postgraduate Programme “Fetal Maternal Medicine’’, Medical School, National and Kapodistrian University of Athens, 11527 Athens, Greece; 2Department of Obstetrics and Gynaecology, Hospital of Langenthal—Region Oberaargau, 4900 Langenthal, Switzerland; 3Second Department of Obstetrics and Gynaecology, “Aretaieion’’ University Hospital, Medical School, National and Kapodistrian University of Athens, 11528 Athens, Greece

**Keywords:** conjoined, twins, triplet pregnancy, ICSI, ultrasound

## Abstract

Conjoined twins represent a rare type of monoamniotic twins. Ultrasound assessment during the first trimester can facilitate the diagnosis, however further assessment by colour Doppler studies, 3D imaging, fetal echocardiography and fetal magnetic resonance imaging (MRI) is usually required in order to determine the specific fetal abnormalities and to guide appropriate pregnancy management. This case report presents a rare case of conjoined twins complicating a dichorionic-diamniotic triplet pregnancy, achieved after intracytoplasmic sperm injection (ICSI) and blastocyst transfer. A 44-year-old woman was referred for chorionicity determination to our Fetal Medicine Centre due to suspicion of conjoined twins in a triplet pregnancy. Ultrasound assessment at 11 weeks demonstrated a dichorionic triplet pregnancy which was also complicated by a rare type of conjoined twins (thoracoomphalopagus) and after a successful embryo reduction a neonate of 2200 g was delivered by caesarean section at term. The accurate diagnosis and early detection of conjoined twins by a fetal medicine specialist is crucial, especially as far as multiple pregnancies with three or more fetuses are concerned.

## 1. Introduction

Multiple pregnancies are generally considered high risk pregnancies as they are associated with various complications and therefore with an increased risk for adverse perinatal outcomes in comparison to singleton gestations. Nowadays, the prevalence of multiple pregnancies has increased due to the rising maternal age and the extended use of assisted reproduction techniques (ART) and twins account for almost 3% of live births. Regarding prevalence of triplets and higher order multiples, it has been reported to be about 87.7 per 100,000 births in the United States in 2019 [1]. Notably, prevalence of triplets and higher order multiples has significantly declined following a peak in 1998 due to the transfer of fewer embryos and selective termination processes [2].

Conjoined twins (CT) constitute a rare and complex complication of monozygotic twinning. The incidence of CT is estimated to be 1.5 per 100,000 births worldwide, which is considerably lower compared to the incidence among on-going pregnancies, due to selective termination of CT pregnancies, higher spontaneous abortion rates, and higher stillbirth incidence among CT pregnancies [3]. This entity of uncertain aetiology, if unnoticed early, implies major complication risks to the pregnancy [4]. Foetuses share a single amniotic cavity and some of their body parts are fused [5]. This is thought to be a result of the separation of the zygote after the 12th day post fertilization [6]. The process by which monozygotic twins form CT is not well described [7].

The occurrence of conjoined twins in a triplet pregnancy is extremely rare and is also associated with high perinatal mortality. Despite their rarity, these pregnancies represent a significant obstetric challenge associated with high perinatal mortality and pose several diagnostic and managing dilemmas [8].

We present a rare case of CT complicating a dichorionic triplet pregnancy, conceived following intracytoplasmic sperm injection (ICSI) and blastocyst transfer, diagnosed antenatally by ultrasonography. A review of similar reports is also presented to illustrate the importance of early prenatal diagnosis and counselling and how it can improve pregnancy outcomes.

## 2. Case Report

A forty-four-year-old gravida 1 para 0 with unremarkable obstetric and gynaecologic history underwent an ICSI procedure due to male infertility. Two blastocysts were transferred and ultrasound examination at 6 weeks revealed a dichorionic diamniotic twin gestation, with two separate and distinct foetal heartbeats.

At 11 weeks an ultrasound reassessment was scheduled, during which a dichorionic-diamniotic triplet pregnancy complicated by a set of conjoined twins was diagnosed. The first amniotic sac contained a single living foetus (triplet A) and foetal size was in accordance with gestational age. In the second amniotic sac, there were two foetuses (triplets B and C) fused from the upper thorax down to the abdomen (thoracoomphalopagus) (Figure 1). In order to offer counselling to the parents regarding postnatal surgical separation options of the conjoined twins their hearts were examined by a foetal cardiologist. Ultrasound cardiac examination showed that the embryos B and C appeared fused at their sternum and their hearts were fused at their atrial level. The heart of embryo C consisted of three chambers with the left sided atrioventricular valve being atretic and the posterior sided ventricle being hypoplastic. The heart of embryo B consisted of four chambers and there was a large ventricular septal defect. It was not possible to visualize the type of the atrioventricular connection. Two intact aortic arches were visible, one for each embryo (Figure 2, Figure 3 and Figure 4). Parents were informed that more discrete anatomic details could be visible after 3–4 weeks and that due to the cardiac fusion and the presence of the three-chambered heart in embryo C, the short-term prognosis for both embryos was not favourable.

Following counselling, the parents decided for selective reduction of CT and an uncomplicated embryoreduction procedure under ultrasound guidance was performed, by intracardiac potassium chloride injection at 11 weeks and 1 day of gestation.

A follow-up ultrasound examination, performed one week later, confirmed a single alive embryo and demise of the conjoined twins. At this stage assessment of the risk for chromosomal abnormalities for the living single foetus was offered to the couple. The crown-rump length (CRL) of embryo A was 51.5 mm, and the nuchal translucency (NT) was measured 1.3 mm. Regarding the rest of first trimester ultrasound markers for chromosomal abnormalities, the nasal bone was present, there was no tricuspid regurgitation and the flow in ductus venosus was normal. Foetal anatomy was normal for the gestational age and the risk for Down’s was 1:461. After counselling the parents decided to avoid invasive testing for chromosomal abnormalities.

During the anomaly scan at 20 weeks and 2 days of gestation, foetal growth was normal and there were no structural defects or any markers for chromosomal abnormalities. Ultrasound assessment at 31 weeks of gestation revealed foetal growth on the 10th centile (1414 g), normal amniotic fluid and foetal Doppler studies. Foetal growth was examined four weeks later showing consistent growth.

The subsequent course of the gestation was uncomplicated and a healthy neonate, weighting 2200 g, was born by caesarean section, at 38 weeks and one day of gestation due to rupture of membranes and parental request. Even though the Apgar scores at the 1st and at the 5th minutes after birth were 8 and 10 respectively, the neonate showed signs of respiratory distress half an hour following delivery and admitted at the neonatal intensive care unit. The neonate was then discharged after ten days.

## 3. Discussion

In this case report we presented a case of CT complicating dichorionic triplet pregnancy after intracytoplasmic sperm injection (ICSI) and blastocysts transfer. The diagnosis was achieved during the 10th week of gestation by ultrasound and after embryoreduction of the CT a healthy neonate was born.

CT represent a rare type of twins, accounting for approximately 1% of monozygotic twins, and are almost always monochorionic monoamniotic. The embryology of this anomaly remains uncertain and there are no specific risk factors that have been consistently associated with this condition. However, a predominance of female cases has been noted as well as a series of various congenital anomalies which are frequently reported in CT, such as genitourinary tract and central nervous system (CNS) malformations [2].

A multicentre worldwide study, including a total of 383 sets of CT obtained from 26,138,837 births reported that the prevalence of CT was 1.47 per 100,000 births in total (95% CI: 1.32–1.62) [2]. In on-going pregnancies, half of the foetuses died in utero while in the rest of the cases, women delivered usually before term (before 32 weeks), following early rupture of membranes [9]. According to other estimations, 40% of CT can be stillborn and one-third can die within 24 h of birth [2]. Prognosis among survivors depends on the type and degree of conjunction and on the presence or absence of associated congenital anomalies. Interestingly the worst prognosis affects CT with sharing liver and/or heart [10,11].

CT are categorized, depending on the fused body part or the site of the fusion as cephalopagus, thoracopagus, omphalopagus, ischiopagus, parapagus, craniopagus, rachipagus, and pygopagus. The most common type appears to be the thoracopagus type showing a 42% predominance [2]. Collins et al., reported that the vast majority of thoracopagus twins are complicated with major congenital heart disease (94.4%) mainly associated with single-ventricle pathology [12]. The outcome of thoracopagus twins with conjoined hearts remains poor due to failure of separating conjoined and single ventricles [13].

Before the implementation of ultrasound in clinical practice, CT were usually diagnosed at delivery usually following dystocia. Tan et al. reported just nine cases of CT in a triplet pregnancy, published before 1971 [13]. The total perinatal mortality rate was estimated 89% for the CT and 56% for the normal triplet. Furthermore, the preterm delivery rate before 32 weeks was 50% [13].

Due to the increasing use of ultrasound, CT can be now diagnosed during the first trimester assessment. Criteria for ultrasonographic diagnosis of CTs include the existence of inseparable bodies, the absence of a separating amniotic membrane, the as well as the absence of change in relative positions on repeated examinations. Other criteria include an unusual proximity of the foetuses and the existence of oligohydramnios which occur in 50–76% of cases. During the anomaly scan at 20 weeks, location and extent of the conjoined foetal structures can be accurately defined and additional imaging studies facilitate further assessment [2,11].

Prenatal ultrasound diagnosis is not always easy for the diagnosis of CT in the first trimester and both false positive and false negative diagnoses have often occurred. Despite the repeated and careful transvaginal ultrasonography, distinction between anatomical parts of the foetuses seems to be difficult. Although first-trimester diagnosis of conjoined twins is now a realistic option, there is a high rate of false-positive diagnosis, when scanning is performed before the 10th week of gestation [8]. Conjoined twins may be misdiagnosed with monoamniotic twins, lymphangioma, teratoma, and/or neoplasm [9].

Apart from two-dimensional (2D) ultrasound, colour Doppler, three-dimensional (3D) ultrasound, foetal echocardiography, and MRI may facilitate CT diagnosis, detailed characterization of the fused part, and associated anatomic anomalies. Colour Doppler usually reveals details of the blood circulation and echocardiography is mandatory in all cases. When necessary, further evaluation with foetal 3D ultrasound can play a vital role to simplify accurate diagnosis of these twins and allow a clearer depiction of their anatomy. 3D ultrasound potentially supplies the clinicians with valuable information for the decision making, regarding the obstetric management, as it enables accurate prenatal visualization of the fusion site and reveal the extent and severity of the malformations [7,12,14,15]. In contrast, Pajkrt and Jauniaux support that 3D imaging does not improve on the diagnosis achieved using 2D ultrasound, as the additional practical medical information is really low, compared to the 11–14 weeks ultrasound examination [9]. MRI should be considered as an alternative for prenatal characterization of structural anomalies, since there is no evidence that MRI is associated with any risk to the developing foetus [10,16]. CT should be referred to specialized centres for detailed foetal anomaly and echocardiographic assessment to evaluate prognosis [17].

With regard to pregnancies achieved with the use of ART, multifoetal multichorionic pregnancies are quite common [18]. These pregnancies require early and repeated ultrasonographic evaluation, to detect implantation, foetal cardiac function, or ectopic pregnancies [19]. Given the opportunity, chorionicity can be diagnosed with accuracy, providing the clinicians with valuable information for the following obstetric care of these pregnancies [20,21,22,23]. Recent data suggest that the rate of monozygotic twinning is increased when using ART, especially when blastocysts are transferred, but the risk for CT has not been evaluated after ART [24,25,26].

Several attempts have been made to identify potential risk factors for CT among pregnancies conceived by ART [27,28,29,30,31]. Hirata et al., reviewed 9 cases of CT, complicating triplet or quadruplet pregnancies conceived after ART [32]. Most of them used assisted hatching or ICSI, indicating that there is a need for more detailed investigation and accurate evaluation of such cases. It has been proposed that, manipulation of the zona pellucida, is a predisposing factor for conjoined twins [32,33,34]. Additionally, delayed blastocyst-stage transport, delayed implantation, in vitro culture condition, culture time, and the manipulation of the egg have been commonly proposed to be associated with monochorionic and conjoined twins [35,36,37]. During hatching, embryos may be trapped in a gap in the zona pellucida [38]. The twins can continue to develop when the inner cell mass splits in half. If the trophoblast does not also split, but remains intact, this results to the formation of monochorionic mono-amniotic twins. If there is also incomplete separation of the inner cell mass, this probably leads to the formation of conjoined twins [20]. Another hypothesis proposes incomplete cleavage of the embryonic disk at 13–15 days after conception as possible mechanism for the formation of CT [2,13]. Finally, incomplete fusion of two embryos could result to a CT phenotype [39]. Despite all these hypotheses regarding CT aetiology, no obvious genetic, environmental or demographic predisposition has been identified [22].

Early diagnosis of CT during the first trimester ultrasound scan allows parental counselling regarding management options, which is important especially in triplet or higher order pregnancies and pregnancies after ART that are considered as “precious baby” pregnancies [8,13]. Selective termination of the CT aims to prevent mortality or serious morbidity of the healthy foetus mainly due to preterm birth [14]. The procedure, however, is associated with a miscarriage rate up to 10% [7]. In case of monochorionic triplet pregnancy, cord occlusion by laser ablation of the single umbilical cord is preferable, compared to intracardiac potassium chloride injection. Placental vascular anastomoses between the normal foetus and the conjoined twins in such cases increase the risk of subsequent death or neurological complications in the survivor foetus [14]. Referral to an experienced centre should follow CT diagnosis. The multidisciplinary team should involve maternal foetal medicine, paediatric surgery, neonatology, and radiology specialists. Since these pregnancies are extremely rare, management is mainly based on clinical experience, expert opinion, and published data from case reports and small series.

Sepulveda et al., has reviewed the literature, in order to identify cases of conjoined twins in triplet pregnancies and investigated the impact of early diagnosis on the prognosis of the non-conjoined triplet [8]. The majority of these cases (6 out of 10) were classified as dichorionic and regarding conjoining the most frequent type was thoracopagus [17]. In order to be in position to offer treatment options, the certainty in the diagnosis of different chorionicity is a precondition. Based on clinical experience, to decrease the possible negative outcome rates, an early decision and early treatment are required. In addition, the gestational age at delivery is greater with early procedures as compared with interventions made later [17]. When selective termination of monochorionic conjoined twins by intracardiac potassium chloride injection into the conjoined twins’ heart is carried out, there is an increased risk for unintended death even of the healthy triplet [19,20]. On the contrary, early selective termination in dichorionic gestations resulted in term delivery of the healthy triplet [7,20]. This is a strong indication that early (preferably first trimester) and accurate determination of chorionicity, is crucial for the decision of the optimum management in such cases [21].

In order to identify other reported cases of CT in triplet pregnancies after ART, we performed an extensive PubMed search of the literature in English. A list of keywords including “conjoined twins,” “triplet pregnancy,” “monochorionic”, “dichorionic diamniotic,” “multiple pregnancy”, “IVF”, “ART” and “ICSI” was used in our search algorithm. The following details have been recorded for each case and are presented in Table 1: country of origin, maternal age, gravity and parity status, chorionicity and amnionicity of triplets, whether selective termination was performed, and pregnancy and neonatal outcomes. To the best of our knowledge, 10 cases of triplet pregnancies with conjoined twins after ART have been previously reported in the literature: five cases with IVF, one with ovulation induction, and four cases with ICSI [14,21,32,40,41,42,43,44,45,46]. Seven out of the ten cases described thoracoomphalopagus CT, 2 omphalopagus CT, and only one case of thoracopagus CT was described. Interestingly, Talebian et al. and Yuan et al. described monochorionic diamniotic (MCDA) CT [43,45].

## 4. Conclusions

In multiple pregnancies, determination of chorionicity and amnionicity is essential for counselling and management. Monochorionic monoamniotic pregnancies should be thoroughly assessed by ultrasound in order to exclude CT. Early detection and accurate diagnosis by experts are of outmost importance. When CT complicate a triplet pregnancy, chorionicity is crucial for the decision on the future obstetric management, as well as the prognosis for the non-conjoined foetus. In case of diagnosis of different chorionicity, a selective reduction of the abnormal pregnancy should be offered. Further research in epidemiology and maternal foetal medicine is necessary to examine the etiologic processes involved and to determine the pathophysiology of this condition.

## Figures and Tables

**Figure 1 children-09-01549-f001:**
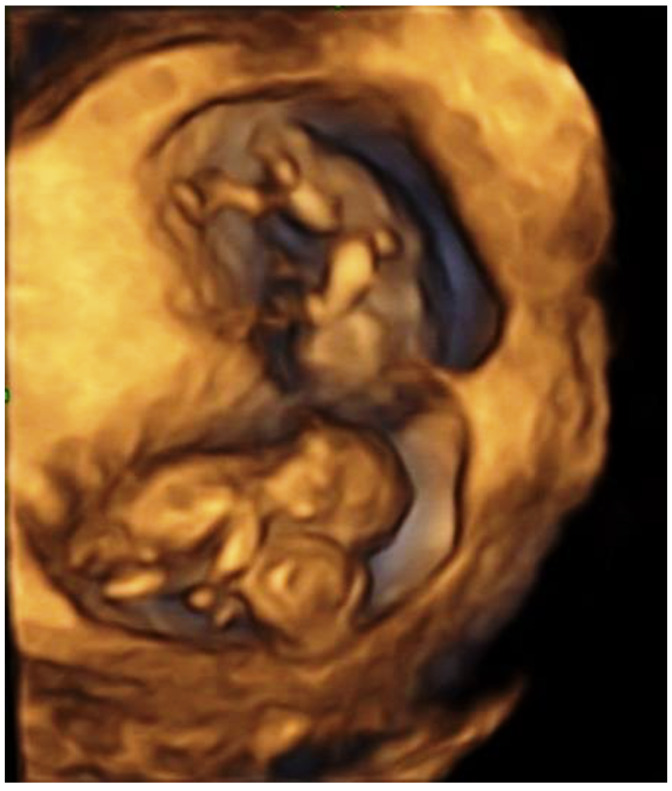
CT in a triplet pregnancy revealed during a 4d ultrasound examination.

**Figure 2 children-09-01549-f002:**
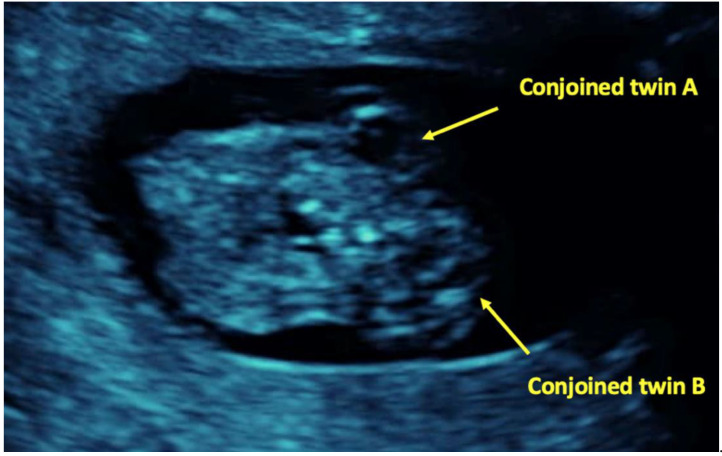
Conjoined twins, longitudinal plane; extent of ventral union: thorax.

**Figure 3 children-09-01549-f003:**
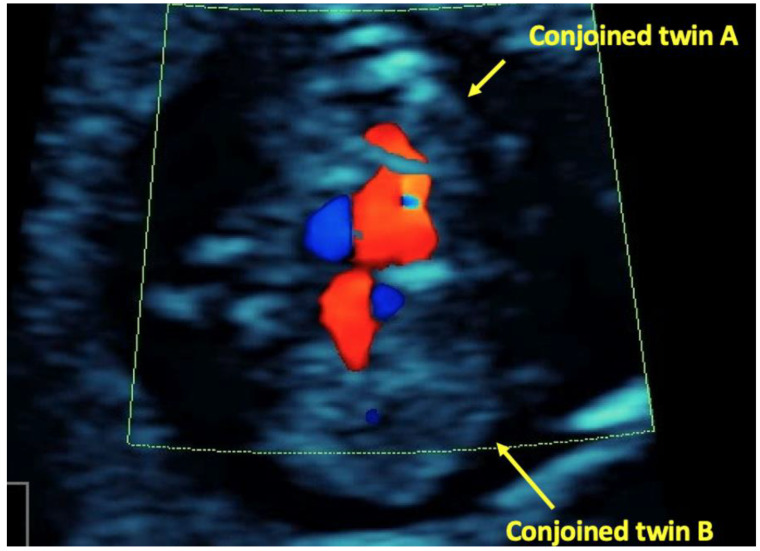
Conjoined twins, transverse plane Color Doppler showing fused cardiac structure.

**Figure 4 children-09-01549-f004:**
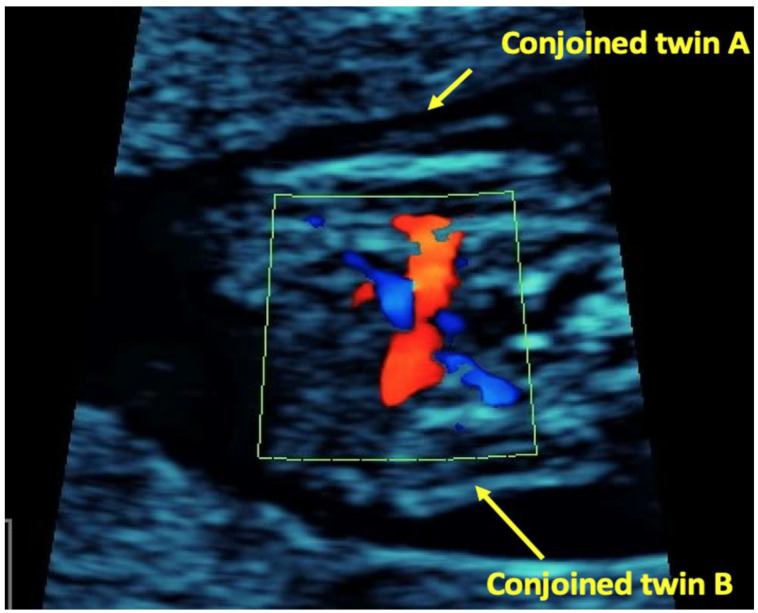
Conjoined twins, longitudinal plane Color Doppler showing fused cardiac structure.

**Table 1 children-09-01549-t001:** Reported cases of triplet pregnancy with conjoined twins after assisted reproduction techniques.

Case No.	Authors	Country	Age	Gravity	Parity	Treatment	Triplet Type	Type of Conjoining	Gestational Age at Diagnosis	Selective Termination	Result	No. of Newborns
1	Skupski et al., 1995 [40]	USA	35	NS	2	IVF	DCDA	Thoraco-omphalopagus	12 weeks	+	Ongoing pregnancy	NA
2	Goldberg et al., 2000 [14]	Israel	28	1	0	ICSI	DCDA	Thoraco-omphalopagus	8 weeks 4 days	+	Ongoing pregnancy	NA
3	Timor-Tritsch et al., 2000 [41]	USA	NS	NS	NS	IVF	DCDA	Omphalopagus	10 weeks	+	NS	1
4	Charles et al., 2005 [42]	Australia	20	NS	NS	IVF	DCDA	Omphalopagus	10 weeks	+	Death due to premature delivery at 21 weeks of gestation	0
5	Hirata et al., 2009 [32]	Japan	34	1	1	ICSI	DCDA	Thoracopagus	8 weeks	-	Spontaneous cardiac arrest of the CTs at 10 weeks and 3 days of gestation A male neonate weighing 2792g was born at 39 weeks of gestation	1
6	Shepherd et al., 2011 [21]	Canada	32	1	0	Ovulation Induction	DCDA	Thoraco-omphalopagus	13 weeks 1 day	+	A male neonate weighing 3590g was delivered vaginally at 40 of weeks gestation	1
7	Talebian et al., 2015 [43]	Iran	38	1	0	ICSI	MCDA	Thoraco-omphalopagus	12 weeks 2 days	+	Selective fetocide performed at 16 weeks of gestation All fetuses were spontaneously aborted	0
8	Castro et al., 2017 [44]	Brazil	32	NS	NS	ICSI	DCDA	Thoraco-omphalopagus	9 weeks	-	Spontaneous cardiac arrest of the CTs at 13 weeks of gestation A female neonate weighing 2750g was vaginally delivered at 38 weeks	1
9	Yuan et al., 2017 [45]	China	39	3	0	IVF	MCDA	Thoraco-omphalopagus	10 weeks	-	Termination of pregnancy by induced abortion	0
10	Liu et al.,.2021 [46]	China	22	1	0	IVF	DCDA	Thoraco-omphalopagus	13 weeks 5 days	+	A female neonate weighing 2760g was delivered by caesarean section at 37 weeks and 4 days of gestation	1
11	Eleftheriades et al., 2022 (our case)	Greece	44	1	0	ICSI	DCDA	Thoraco-omphalopagus	11 weeks	+	A healthy neonate weighting 2200g was delivered by caesarean section at 38 weeks and 1 day of gestation	1

NS: non specified, NA: non applicable, MCDA: monochorionic diamniotic, DCDA: dichorionic diamniotic, IVF: in vitro fertilization, ICSI: intracytoplasmic sperm injection.

## Data Availability

The authors confirm that the data supporting the findings of this study are available within the article.

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
