# Peer review of "Conjoined Twins Complicating a Dichorionic Triplet Pregnancy after Intracytoplasmic Sperm Injection: A Case Report and Review of the Literature"

_children, 2022, doi:10.3390/children9101549_

Round 1

Reviewer 1 Report

This is an interesting case about conjoined twin, as the author report, the case was diagnosed based on an ultrasound examination showing that embryos B and C appeared fused at their sternum and their hearts were fused at their atrial level. But figure 1 does not give us the information.  The author should present a detailed image of the diagnosis mentioned above.

Author Response

Reviewer 1

This is an interesting case about conjoined twin, as the author report, the case was diagnosed based on an ultrasound examination showing that embryos B and C appeared fused at their sternum and their hearts were fused at their atrial level. But figure 1 does not give us the information.  The author should present a detailed image of the diagnosis mentioned above.

Answer: We thank the reviewer for their comment but unfortunately, due to technical problems we couldn’t find other saved images depicting CT and in particular their heart anatomy.

Reviewer 2 Report

This is a case report of dichorionic-diamniotic triplets complicated by conjoined twins in a monochorionic sac after ISCI procedure. Additionally the authors made relevant literature review on the CT and triplet pregnancies. It is a valuable description of the very rare clinical situation with the proposal of modern approach. I find it very useful in daily practice of a fetal specialist.

1/ Replace "mini-review" with just "review" in your title.

2/ I cannot find the table 1 you mention about.

3/ Specify why you did not offer cell free fetal DNA testing based on NGS e.g. SNP to check for genetic conditions and zygosity of the triplets and conjoined twins (CT).

4/ What was the gender of CT in your case? Was it the same as the survivor? You mentioned that females as more predominant for being affected by CT.

5/ Was it monozygotic or dizygotic CT? What is the mechanism for MZ vs DZ conjoining? It would be very interesting and educative for readers.

6/ You may want to present as a graph different types of CT based on proposed mechanisms in mono vs dizygotic and mono vs dichorionic multiple pregnancies.

7/ Why you chose potassium chloride and not cord occlusion (e.g. laser ablation or radiofrequency) for selective termination? You mentioned about possible vessel connections even in dichorionic pregnancies

Thank you

Reviewer 3 Report

It is a well-written, interesting case report and review of a relevant literature. The case is very radÄ™ and it is important to publish such studies in order to improve prenatal diagnostics and managemng.

Round 2

Reviewer 1 Report

This is a case report about conjoined twins diagnosed at early pregnancy but the authors can not provide the images for the diagnosis they describe as " embryos B and C appeared fused at their sternum  and their hearts were fused at their atrial level. The heart of embryo C consisted of three chambers with the left sided atrioventricular valve being atretic and the posterior sided ventricle being hypoplastic. The heart of embryo B consisted of four chambers and there was a large ventricular septal defect. Those description in the manuscript should be deleted.

Round 3

Reviewer 1 Report

The authors had responded me well.